# Disclosure of study funding and author conflicts of interest in press releases and the news: a retrospective content analysis with two cohorts

Petroc Sumner [1], Lisa Schwartz,[2] Steven Woloshin,[2,3] Luke Bratton,[1] Christopher Chambers[1]

¹School of Psychology, Cardiff University, Cardiff, UK
²Center for Medicine and the Media, The Dartmouth Institute for Health Policy and Clinical Practice, Lebanon, New Hampshire, USA
³The Lisa Schwartz Foundation for Truth in Medicine, Norwich, Vermont, USA

**Correspondence to**
Dr Petroc Sumner;
SumnerP@cardiff.ac.uk

## ABSTRACT

**Objectives** To examine how often study funding and author conflicts of interest are stated in science and health press releases and in corresponding news; and whether disclosure in press releases is associated with disclosure in news. Second, to specifically examine disclosure rates in industry-funded studies.

**Design** Retrospective content analysis with two cohorts.

**Setting** Press releases about health, psychology or neuroscience research from research universities and journals from 2011 (n=996) and 2015 (n=254) and their associated news stories (n=1250 and 578).

**Primary outcome measure** Mention of study funding and author conflicts of interest.

**Results** In our 2011 cohort, funding was reported in 94% (934/996) of journal articles, 29% (284/996) of press releases and 9% (112/1250) of news. The corresponding figures for 2015 were: 84% (214/254), 52% (131/254) and 10% (58/578). A similar pattern was seen for the industry funding subset. If the press release reported study funding, news was more likely to: 22% if in the press release versus 7% if not in the press release (2011), relative risk (RR) 3.1 (95% CI 2.1 to 4.3); for 2015, corresponding figures were 16% versus 2%, RR 6.8 (95% CI 2.2 to 17). In journal articles, 27% and 22% reported a conflict of interest, while less than 2% of press releases or news ever mentioned these.

**Conclusions** Press releases and associated news did not frequently state funding sources or conflicts of interest. Funding information in press releases was associated with such information in news. Given converging evidence that news draws on press release content, including statements of funding and conflicts of interest in press releases may lead to increased reporting in news.

## INTRODUCTION

Medical journals, funders and academic institutions routinely call on researchers to disclose funding sources and financial conflicts of interest. Doing so is designed to increase the trustworthiness of the research process and allows readers to decide whether industry entanglements merit heightened scepticism when interpreting results.

### Strengths and limitations of this study

► Reporting of study funding and conflicts of interest was assessed using a large cohort of press releases (1250) and news (1828) across two cohorts from separate years.
► The association between news and press release reporting was also assessed.
► The study is correlational and retrospective.
► The data included mainstream newspapers and internet media, but not broadcast media or social media.
► Generalisability to other countries and languages is unknown.

There are no corresponding disclosure requirements for research reported in the lay press: only 3% of the largest circulation US newspapers had an explicit policy about reporting industry funding of medical research.[1] Published reports have documented substantial under-reporting of author conflicts of interest and industry funding in the lay media.[2–4] Such under-reporting matters since many people—including physicians[5]—learn about the results of medical research from the news.[6]

The majority of news stories about news health-related discoveries are stimulated by press releases from universities or academic journals. Several studies suggest that press releases may strongly influence the content of subsequent media coverage. For example, news stories were more likely to report absolute risks, intervention harms and study limitations when they were reported in the medical journal press release.[4] Similarly, other aspects of news reports appear strongly associated with the wording and information in corresponding press releases, such as making causal claims from correlational data, exaggerating the relevance to humans

of animal research, 'spin', or caveats to mitigate such exaggeration.[7–13]

We analysed how often funding and conflicts of interest are mentioned in biomedical and health news stories and their corresponding journal and institution press releases. We examined whether the presence of such statements in press releases is associated with their presence in news. We then specifically examine the subset of studies that had industry funding.

## METHODS
### Study design
We scrutinised two collections of health-related news stories, press releases and associated journal articles for reports of funding and conflicts of interest. We analysed the reporting frequencies and the association between reports in news and press releases.

### Source materials
The first database contains 1250 news stories, 996 press releases and 996 associated journal articles.[8 9] This database was collated by selecting all the press releases related to human health published throughout 2011 from eight leading international biomedical journals (*Lancet*, *British Medical Journal* (*BMJ*), *Science*, *Nature*, *Nature Neuroscience*, *Nature Immunology*, *Nature Medicine* and *Nature Genetics*) and 20 leading UK universities (The Russell Group; see figure 1). The corresponding journal article for each press release was sourced, as were subsequent news stories in mainstream print and internet outlets.[8 9] The second database[12 13] contains 578 news stories, 254 press releases and 254 associated journal articles. This was collated by selecting press releases related to human health published between January and June 2015, from 26 UK universities (including the Russell Group and additional universities

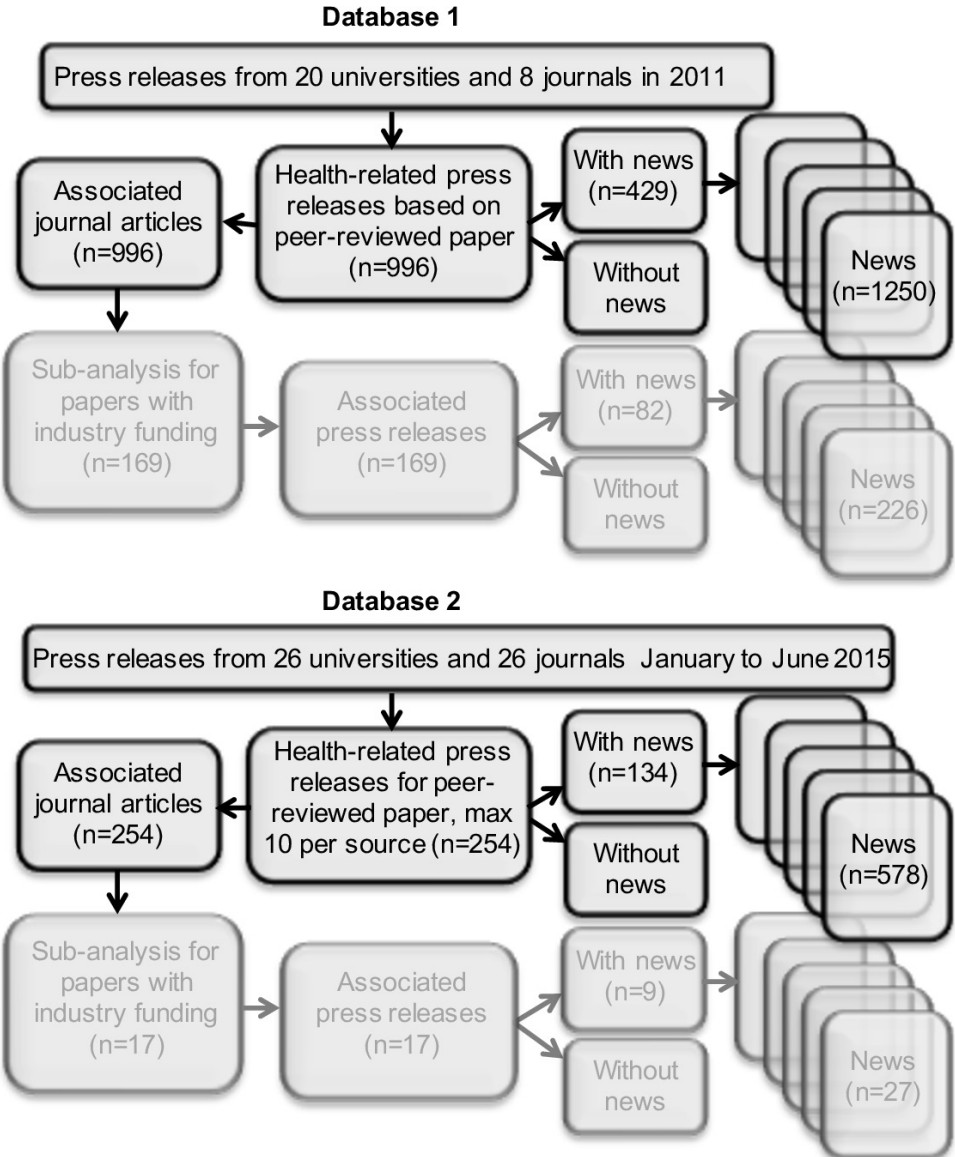

**Figure 1** Flow diagram describing the two datasets and the available numbers for analysis.

in Adams *et al*[12]) and 26 journals (10 journals affiliated with the *BMJ* group, 16 with the BMC group).

## Search methods and inclusion criteria

Press releases were identified from publicly available repositories (web pages or EurekAlert) or non-public sites for journalists (Nature Publishing Group provided us with free access to all their press releases). The inclusion criteria were: health-related topic (broadly defined to include all biomedical sciences, diet, lifestyle, psychology) based on a peer-reviewed published journal article (that we could access). In the 2011 set all eligible press releases were included. In the 2015 set, the contribution from each institution had been capped to 10 press releases, selected randomly if more than 10 were available (for feasibility reasons[12 13]). If two press releases for the same journal article were identified (one from the university and one from the journal), only one was used, randomly selected. To identify any print and online news stories related to each press release, we searched Lexis-Nexis, BBC.co.uk, uk.reuters.com and Google with keywords up to 28 days after publication of the press release, and up to 1 week before (to allow for potential news before the embargo was lifted).

## Data extraction and coding

Journal articles, press releases and related news coverage were coded by research assistants using a prespecified protocol, to extract information about study funding and the authors' reported conflicts of interest. Study funding was coded as industry (eg, GlaxoSmithKline, Pfizer, etc), government (eg, the research councils such as the National Institutes of Health and US National Cancer Institute, Medical Research Council (MRC), Biotechnology and Biological Sciences Research Council (BBSRC), Economic and Social Research Council (ESRC), etc), charity (eg, Wellcome Trust, Cancer Research UK, the British Heart Foundation, etc), internal/other (eg, self-funded or university-funded) or none mentioned. The first-mentioned source was always coded, and then industry funding if it was listed. Therefore studies could belong to more than one category (eg, government and industry). For the specific analysis of industry funding, a study was included if an industry source was mentioned, regardless of position in the list of funders. Coders located the 'Conflict of interest (COI)' or 'Competing interests' sections of the article and determined whether there was no declaration found, a declaration of no COI (eg, 'The authors declare no COI') or whether any author declared conflicts of interest (eg, 'Author X is a paid consultant to Y company').

The press releases and news stories were simply coded for whether funding or conflicts of interest were reported, and whether the press release and news specifically mentioned any industry funding.

The raw materials and protocols for the two databases are available at https://figshare.com/articles/dataset/InSciOut/903704 and https://osf.io/apc6d/. The latter also contains the extracted data used for this study, in the folder 'Processed data/funding and conflicts of interest'.

## Coding reliability

For the 2011 set, a second research assistant independently coded a randomly selected sample of 28% of press releases and associated news (23% of total news stories). Observed agreement was 94% for coding the type of funding source, 92% for whether press release (PR) reported funding, 94% for whether news reported funding, 98% for the study's COI statement and 99% for both whether press release and news mentioned COI (we do not calculate kappa as it is unreliable when agreement is this high). For coding disagreements, one answer was randomly selected. For the 2015 set, a second research assistant independently coded all texts, and any discrepancies were subsequently highlighted and discussed to reach a consensus conclusion. A third research assistant arbitrated if disagreements remained (very rare).

## Analysis

We first report analyses of all journal articles and associated press releases descriptively, followed by descriptive analysis of news. These are separated by year to illustrate the natural range of fluctuation, rather than to examine trends with time (the differences in sampling would undermine such analysis). Descriptive analyses were done in Stata V.14.2. Since the main association analysed—relating the mention of funding in press releases to news stories—is limited to the press releases with media coverage, we also give descriptive information for these subsets in table 1. To examine the relationship between news and press releases, we used generalised estimating equations to account for clustering of news stories for each press release (using an exchangeable working correlation; in SPSS, V.23). For conflicts of interest, the association between press releases and news could not be analysed because so few news stories mentioned conflicts of interest.

## Patient and public involvement

No patients or participants were involved in this study.

## RESULTS

### Disclosure of study funding and conflicts of interest in the journal article

Among all 996 studies in 2011, 94% (934) listed sources of funding in the journal article and 17% (169) reported industry funding. The corresponding figures for 2015, among 254 studies, were 84% (214) and 7% (17), respectively. In about one-quarter of studies (27% and 22% for 2011 and 2015), one or more authors declared a COI (see table 1 for all numbers and %).

### Disclosure of study funding and conflicts of interest in the press release

Press releases reported a funding source 29% and 52% of the time (respectively, for 2011 and 2015). Press releases

**Table 1** Frequency of funding sources and conflicts of interest in journal articles, press releases and news reports

| | All studies | | Studies with media coverage | |
|---|---|---|---|---|
| | **2011** | **2015** | **2011** | **2015** |
| **Information in journal article** | **N=996** | **N=254** | **N=429** | **N=134** |
| Funding source reported | | | | |
| Any funding reported | 94% (934/996) | 84% (214/254) | 93% (401/429) | 82% (110/134) |
| Any industry | 17% (169/996) | 7% (17/254) | 19% (82/429) | 7% (9/134) |
| Single non-industry sources | | | | |
| Government | 6% (56/996) | 28% (71/254) | 6% (24/429) | 22% (30/134) |
| Charity | 13% (125/996) | 6% (16/254) | 13% (55/429) | 8% (11/134) |
| Internal/other | 4% (38/996) | 8% (20/254) | 4% (18/429) | 8% (11/134) |
| Multiple non-industry sources | 55% (546/996) | 41% (104/254) | 52% (222/429) | 42% (56/134) |
| None stated | 6% (62/996) | 16% (40/254) | 7% (28/429) | 18% (24/134) |
| Authors COI disclosed | | | | |
| Declare 'none' | 57% (563/996) | 50% (126/254) | 54% (231/429) | 48% (64/134) |
| Declare >1 conflict | 27% (268/996) | 22% (55/254) | 29% (123/429) | 21% (28/134) |
| No statement | 16% (165/996) | 29% (73/254) | 17% (75/429) | 31% (42/134) |
| Information in press releases | | | | |
| Funding source reported | | | | |
| Report any funding source | | | | |
| All press releases | 29% (284/996) | 52% (131/254) | 35% (150/429) | 57% (76/134) |
| University | 59% (253/426) | – | 62% (127/206) | – |
| Journal | 5% (31/570) | – | 10% (23/223) | – |
| Report industry funding as % of studies with industry funding | 14% (24/169) | 41% (7/17) | 24% (20/82) | 44% (4/9) |
| Authors COI disclosed | | | | |
| Report COI as % of studies where COI declared | 0.5% (1/268) | 2% (1/55) | 0% (0/123) | 4% (1/28) |
| Report no COI as % of studies that declared none | 1% (4/563) | 0% (0/126) | 0.4% (1/231) | 0% (0/64) |
| Information in news stories | | | | |
| Funding source reported | | | | |
| Report any funding | – | – | 9% (112/1250) | 10% (58/578) |
| Report industry funding as % of studies with industry funding | – | – | 17% (38/226) | 0% (0/27) |
| Authors COI disclosed | | | | |
| Report COI as % of studies where COI declared | – | – | 1% (4/380) | 0% (0/114) |
| Report no COI as % of studies that declared none | – | – | 0.1% (1/675) | 0% (0/286) |

Studies could belong to more than one funding category (eg, government and industry).
COI, conflict of interest.

specifically mentioned industry funding when promoting industry-funded studies 14% and 41% of the time, respectively. In the larger sample (2011 cohort) we could divide press releases issued by universities from those issued by journals; the universities were more likely to mention a funding source (59% vs 5%; absolute difference=54%, 95% CI 49% to 59%). Reporting of conflicts of interest was rarer: 0% and 2% of press releases (for 2011 and 2015) mentioned a COI where one was declared in the journal article. Reporting of no conflicts was similarly rare: 0.4% and 0% of press releases explicitly reported no conflict for studies that explicitly declared none.

### Disclosure of study funding and conflicts of interest in news stories

For the set of studies with media coverage, reporting of funding sources in news stories was low: 9% for all studies; 17% for industry-funded studies in 2011; 10% for all studies and 0% for industry-funded studies in 2015. Reporting of conflicts of interest was even rarer:

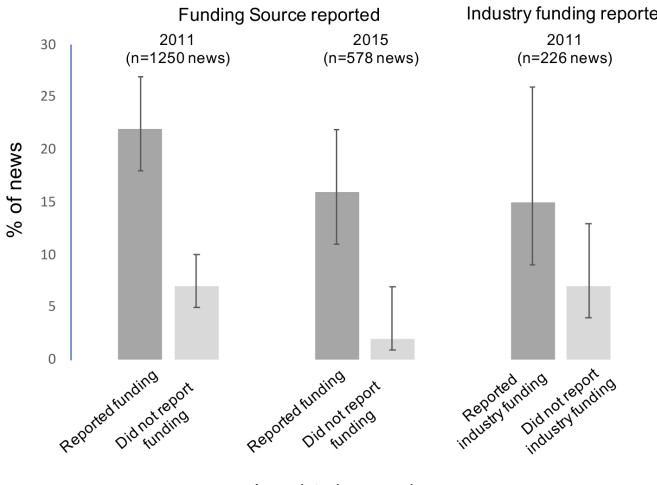

**Figure 2** Proportion of news stories reporting study funding source or industry funding according to whether the associated press release reported the funding source or industry funding. There were zero mentions of industry funding in news in 2015 dataset, so analysis not performed. Error bars are 95% CI. The relative risks for the three plots are: 3.1 (95% CI 2.1 to 4.3); 6.8 (95% CI 2.2 to 17); 2.1 (95% CI 0.94 to 4.5).

1% and 0% of news stories (for 2011 and 2015) reported a conflict in the studies where a conflict was declared in the journal article, while 0.1% and 0% of news explicitly reported no conflict for studies that explicitly declared none.

### Relationship of funding source in press release and the news

If the press release reported a funding source, associated news stories were more likely to report it (figure 2). For the 2011 cohort, 22% of news stories reported funding if in the press release versus 7% if not in the press release; relative risk (RR) 3.1 (95% CI 2.1 to 4.3); absolute difference 15% (95% CI 8% to 23%). For the 2015 cohort, 16% of news stories reported funding if in the press release versus 2% if not in the press release, RR 6.8 (95% CI 2.2 to 17). The results were similar among the subset of 226 news from industry funded studies in 2011: 15% of news stories reported industry funding if in the press release versus 7% if not in the press release; RR=2.1 (95% CI 0.94 to 4.5); absolute difference 8% (95% CI 0% to 18%). For 2015, there were no reports in news of industry funding from the subset of industry funded studies (n=9 studies, 27 news; see table 1).

### DISCUSSION

Our study highlights that reporting of funding sources is not high in either news or press releases from major biomedical journals and leading UK research universities. Neither was industry funding mentioned in the majority of news or press releases based on studies with industry funding. Mentioning conflicts of interest—or stating that there were none—was almost vanishingly rare.

Consistent with prior work,[4 8–13] we observed that information—in this case funding source—is more likely to appear in the news when it is noted in the press release. Given that press releases are used as sources for news, we believe this correlation is likely to contain a causal element. In turn, this would provide a means to increase the frequency with which news mentions funding and conflicts of interest, should authors and institutions wish to do so (or develop a policy to do so).

Disclosure of study funding and author conflicts interests matters: non-disclosure may undermine public and professional trust in the integrity of the research, while disclosure is designed to allow readers to approach findings with appropriate scepticism. In an era of mass information with varying credibility, it is particularly important for science and health research to be trustworthy.

Disclosure can only be effective if it reaches readers, most of whom—including many physicians[5]—learn about new research in the lay press. The level of under-reporting that we observed may reflect the lack of explicit media policies about reporting industry funding.[1] We hope that this could change. It could be beneficial if press offices at medical journals, funders and academic institutions were to routinely highlight funding and disclosures in their press releases. One way to routinely operationalise this approach would be to add standard headers in press releases for funding and conflicts of interest as is done in many medical journals. Formal testing of alternate content and formats would inform the creation of more effective press releases. If press releases were made openly available and linked to publications for peers to scrutinise, this might remind authors to declare their conflicts of interest.

A strength of this study is the large datasets of over 1200 press releases, 1200 journal articles and 1800 news when taken together. Several study limitations should be acknowledged. First, since the association between press releases and news stories is observational, we cannot prove causation. Second, we do not infer anything from the fluctuations between years. We were not attempting to analyse trends with time, because the two databases have some differences in sampling method that could confound such analysis. Rather the two cohorts simply illustrate the range of results from different samples. Third, although we searched multiple databases attempting to target all major print and online news outlets, we did not include broadcast media, and we may have missed some media coverage. Fourth, the extent of generalisability is uncertain; while press releases for the leading UK academic universities and many leading journals were covered, and there is no reason to suspect major differences between countries,[9 10] or non-included journals, we cannot rule out that countries differ or that some journals or universities may have different press release policies, nor can we be sure that this relationship is generalisable to all other media, such as social media. Fifth, statements about conflicts of interest in journal articles tend to focus on potential financial interests; non-financial interests can arise that are not stated,[14–16]

and thus not analysed here. For example, belonging to a professional organisation or a research network or consortium can potentially result in entrenched viewpoints, while competition and reward structures within academia can also result in conflicts of interests. Finally, we simplified our coding to whether funding or conflicts were present or absent in press releases and news, and did not capture whether reporting fairly represented the entire set of funding or COI in the study.

We additionally observed that university press releases mentioned funding many more times than journal press releases did (in the 2011 cohort where we could analyse this, see table 1). This difference deserves explanation, but we can only speculate. We believe authors and universities feel obliged (and are sometimes explicitly obliged) to acknowledge their funders—without whom the research could not have taken place. It is also likely that mentioning funders lends authority (to get funding, research projects must normally win a highly selective competition). Journals have their own selective processes for publication, and appear not to feel the need to mention funders, either to acknowledge them or to enhance authority. We hope that journals will adopt policies to highlight funding and conflicts of interest in their press releases.

In conclusion, we believe the research community's commitment to disclosing funding and conflicts of interest should extend to press releases—the most direct way that researchers communicate with the media. This does not seem to be the norm in most press releases issued by academic institutions and journals (at least in 2011 and 2015). It is likely that including such information in press releases would raise the rate is it reported in news.

**Acknowledgements** This article is dedicated to the memory of Lisa Schwartz (1963–2018). LS inspired this investigation and produced the first version of the manuscript. LS died in November 2018. We believe this paper stays true to her spirit and intentions. We thank the following for data collection and related projects that informed our approach and interpretation here: Rachel Adams, Caitlin Argument, Amy Barrington, Jacky Boivin, Lewis Bott, Laura Benjamin, Hannah Coulson, Eleanor Corney, Bethan Dalton, Aimee Davies, Cecily Donnelly, Cameron Dunlop, Rebecca Emerson, Rose Fisher, Oliver Gray, Bethan Hughes, Katie John, Laura Jones, Sarah Mann, Olivia Manship, Hannah Maynard, Hannah McCarthy, Jack Ogden, Amy Parfitt, Naomi Scott, Lauren Stead, Christos Venetis, Solveiga Vivian-Griffiths, Eliza Walwyn-Jones, Claire Weeks, Leanne Whelan, Andrew Williams, Joe Wilton.

**Contributors** PS, LS, SW and CC designed the study based on LS's suggestion. PS and CC collated the 2011 data and LB coded and collated the 2015 data. LS and PS did the analyses. LS wrote the first draft and PS wrote subsequent drafts with key contributions from SW, LB and CC.

**Funding** This project was supported by ESRC grant ES/M000664/1 and Wellcome grant 104943/Z/14/Z. The funders had no role in the analysis or writing of this paper.

**Competing interests** None declared.

**Patient consent for publication** Not required.

**Provenance and peer review** Not commissioned; externally peer reviewed.

**Data availability statement** Data are available in a public, open access repository at https://osf.io/apc6d/.

**ORCID iD**
Petroc Sumner http://orcid.org/0000-0002-0536-0510

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
