## [Reviewer comments · BMJ Open]

ARTICLE DETAILS

TITLE (PROVISIONAL)	Disclosure of study funding and author conflicts of interest in press releases and the news: A retrospective content analysis with two cohorts.
AUTHORS	Sumner, Petroc; Schwartz, Lisa; Woloshin, Steven; Bratton, Luke; Chambers, Christopher

VERSION 1 – REVIEW

REVIEWER	Joel Lexchin York University
REVIEW RETURNED	21-Jun-2020

GENERAL COMMENTS	Thank you for the opportunity to review this study that looked at how often study funding and author conflicts of interest are stated in science and health press releases and in corresponding news; and whether disclosure in news is associated with disclosure in press releases. The results should not be surprising given the findings in the literature cited by the authors and other studies going back at least as far as the one by Moynihan in 2000 that appeared in NEJM. 1. In the Introduction the authors give some reasons for why funding sources and COI should be reported. However, is there any hard evidence that reporting financial conflicts of interest promotes public trust in research? Do readers actually change their minds about the merits of research results when conflicts are reported?2. Are the authors able to break down their results for news media by whether the particular media did or did not have a dedicated health reporter?3. Line 117: Do the authors mean that all authors declared ≥ 1 COI or that one or more authors declared COI.4. Another limitation is that the authors did not search for news stories that were just on broadcast media.
--

REVIEWER	Assem M. Khamis Hull York Medical School, University of Hull, UK
REVIEW RETURNED	09-Jul-2020

GENERAL COMMENTS	The article covers a critical topic -especially at the current time-: to examine the reporting of funding and conflicts of interest in the corresponding news of journal articles and press releases. The topic is crucial because the news and media formulate public knowledge and awareness about the new scientific evidence, and lack of reporting both COI and funding could affect how the public perceives the news and result in bias. The study is shedding the light on the importance of the industry funding not only for the scientific community but also for the lay audience. According to the study
--

	findings, there is a huge lack in reporting of funding sources and authors' conflicts of interest in the news and press releases corresponding to the published journal articles. Major revisions: Page 4 – 7: general comments on the 'Methods' section: 1- It would be helpful for the readers if subheadings were provided: study design, definitions, sources of documents, search methods, inclusion criteria, data extraction and coding, and data analysis. 2- I think there is a repetition in reporting on the numbers of articles, press, and news reported in the methods and results section. It might be better to keep it just in the results section except for the coding reliability statistics. 3- The authors did an excellent job in coding in duplicate and independently to minimize data extraction errors. 4- A flow diagram for the included documents will be helpful to track the documents' selection process and analyses. Similar to the previous study with the documents database https://www.bmj.com/content/349/bmj.g7015 Page 9 – 11: general comments on the 'Discussion' section: 1- It would be helpful for the readers if subheadings were provided: summary of findings, strengths and limitations, comparison to other studies, impact on future research and practices in news and press releases, and conclusions. 2- 'what's next?' it would be great if the authors could reflect how their study findings could help in improving the press and news environment to report on COI and funding in scientific related materials. Should these media sources acquire COI and funding policies to report on the scientific findings? Should the universities have their policies for the press releases? Page 9, line 164: the paragraph started with "We additionally observed that university press releases...." the authors introduced new information that has no prior been mentioned in the methods or the results section. Despite the thoughtful reflection about this observation, the authors are free to provide data on it in the results section to allow the readers to reflect on these numbers or remove this paragraph from the discussion as it will not affect the discussion flow. Minor revisions: Page 2, line 5: Objectives: there is a repetition "in" Table 1 is a little bit crowded with numbers. I would suggest doing the following: 1- 'n'= 996 should 'N' for the baseline denominators for each year 2- I think there is no need to mention 'N' in every statistic and it would be less crowded if n (%) if reported. For example: "Any industry" 169 (17.0) 3- In the row of "All press releases": it is better to stick to column percentages for university and journal instead of row percentages. For example: 253/284 or 253/996. And 'n' for university and journal can be added beside them as follow: University (n= 426) Journal (n= 570) 4- In case of new denominators, N can be reported against the main variable. For example: "Authors conflict of interest disclosed" n= 268
REVIEWER	Andreas Lundh Centre for Evidence-Based Medicine Odense (CEBMO), Odense

	University Hospital, Denmark
REVIEW RETURNED	31-Jul-2020

GENERAL COMMENTS	General comments The authors report on a study of 1250 press releases on scientific articles and corresponding uptake by news media. The authors have previously used this sample to study exaggeration in news and the focus of the current study is whether study funding and authors' conflicts of interest is reported in news. The authors find that study funding is often not reported and reporting of funding in press releases is associated with corresponding reporting in news. In addition, authors' conflicts of interest are rarely reported in both press releases and news media. The study is both interesting and generally well reported. However, I have some comments that the authors may consider. Major compulsory revision None. Minor compulsory revision Title and abstract The title describes the study as a "retrospective observational study" and the abstract as a "retrospective quantitative content analysis". I suggest the authors describe the study using the same terminology throughout the paper and "retrospective observational study" seems somewhat too vague. Since the authors report on two cohorts of journal articles with corresponding press releases and follows these articles over time to see how they are cited in news I think they could also call this a cohort study of two cohorts. p6 Methods A protocol is mentioned in the methods section. I suggest that the authors add the protocol as supplementary information or link to a public available version. p6 Methods Both cohorts include press releases sampled from both journals and universities. However, a single research articles may have multiple press releases e.g. one from the journal and one from each of the co-authors' institutions. It is not clear from the manuscript if the same journal article could be included multiple times with different press releases or if this was not the case how the authors decided which press release to use in case of multiple press releases identified for a single article. p7 para 3 line 1-4 + Table 1 It seems that the category "industry-funded" was used if a study had received any industry funding, despite for example NIH funding being mentioned first. However, according to table 1 in the 2015 cohort of all studies the total is 254, but according to funding categories the sum is 268 studies and for the industry subsample the total is 134 and the sum is 141. The authors should please clarify this discrepancy. p8 para 2 line 1-5 It seems that the authors only coded whether a study with reported funding or conflicts of interest also had funding or conflicts of interest reported in press releases and news and not whether the subsequent reporting adequately reflected the information in the journal article. For example, the journal article might mention multiple funding sources and authors' conflicts of
--

	interest, but only some of these may be reported in press releases and news. The authors should consider discussing this as part of the study limitations. p9 para 2 line 6 It is unclear how the authors converted odds ratios to relative risks. Discretionary revision Abstract objectives and main manuscript The authors describe one of their objectives as “whether disclosure in news is associated with disclosure in press releases”. To better fit with the possible causal nature and temporal relationship I suggest they turn the description of the association around i.e. whether disclosure in press releases is associated with disclosure in news. Abstract results The authors write that the results from the 2011 and 2015 cohorts were similar. However, funding was reported in 52% of 2015 press releases compared with 29% of 2011 press releases. Since this is a result I suggest the authors do not make any qualitative interpretation and delete “similar”. p5 para 3 line 7-8 The last sentence is somewhat unclear. Why not use “are associated with”? p6 para 2 line 11 The authors mention that cohort 1 is from UK universities, but the geographical location of the universities in cohort 2 is not described. p7 para 2 line 8 It is unclear how the “largest” funding source was decided as specific monetary amounts are rarely reported in journal articles. p8 para 2 line 1-3 The coding of funding information from press releases and news is also mentioned in the section “funding – general” above so this seems somewhat like repetition. p8 para 3 For cohort 1 it is not described what the authors did when coders disagreed. p9 results I suggest also writing actual numbers in addition to percentages, e.g. 934 (94%) listed sources of funding..... p9 para 3-4 (heading) I suggest writing coi in full. p10 para 1 The authors describe that press releases rarely reported no conflicts of interest when none were reported in the journal article. However, the subsequent example is not related to authors’ conflicts of interest, but whether the study was independently conducted without funder involvement. A study might be “independently” conducted by authors with personal financial conflicts of interest with the funder. I suggest the authors should provide another example. p11 para 2 line 5-8 I suggest moving this section from the principal finding to the study interpretation section below. P13 para 2 line 12-16 I suggest using the term non-financial conflicts of interest or non-financial interest and consider citing the discussion
--	--

	concerning such interest by Bero and Grundy (Bero PLoS Biol 2016). Table 1 In main text funding source category is internal/other, here it is internal.
--	---

VERSION 1 – AUTHOR RESPONSE

Reviewer: 1

Reviewer Name: Joel Lexchin

Institution and Country: York University

Please state any competing interests or state 'None declared': None declared

Please leave your comments for the authors below

Thank you for the opportunity to review this study that looked at how often study funding and author conflicts of interest are stated in science and health press releases and in corresponding news; and whether disclosure in news is associated with disclosure in press releases. The results should not be surprising given the findings in the literature cited by the authors and other studies going back at least as far as the one by Moynihan in 2000 that appeared in NEJM.

Thank you for pointing out this reference, which is now included as reference 2.

1. In the Introduction the authors give some reasons for why funding sources and COI should be reported. However, is there any hard evidence that reporting financial conflicts of interest promotes public trust in research? Do readers actually change their minds about the merits of research results when conflicts are reported?

We agree that this was carelessly worded in the introduction. We have rephrased it to read: *Doing so is designed to increase the trustworthiness of the research process, allowing readers to decide whether industry entanglements merit heightened skepticism when interpreting results.* The key change in our text is from 'promoting public trust' to 'increasing trustworthiness'. The aim of disclosures is not simply to promote trust; sometimes it might be appropriate for a reader not to trust the source. Rather it is to increase the information available to readers for them to judge trustworthiness.

The evidence on whether disclosure of COI actually changes trust or behaviour one way or another is rather mixed and focuses mainly on the effects of individual cases of disclosure for advisors and advisees, rather than what a norm of routine disclosure might achieve, e.g. <https://journalofethics.ama-assn.org/article/do-conflict-interest-disclosures-facilitate-public-trust/2020-03>

2. Are the authors able to break down their results for news media by whether the particular media did or did not have a dedicated health reporter?

We agree this would be interesting, but unfortunately, we do not have the information needed to do this. Also, given the relatively rare reporting of funding and conflicts in news, the numbers in most categories would not allow a meaningful comparison between the different media outlets.

3. Line 117: Do the authors mean that all authors declared ≥ 1 COI or that one or more authors declared COI.

Thanks for pointing out the ambiguity. We meant the latter, and have clarified the sentence.

4. Another limitation is that the authors did not search for news stories that were just on broadcast media.

We agree, and have added this limitation to the discussion and 'strengths and limitations' bullet points. We did not have access to broadcast media resources at the time of this data collection, but we corrected this for our subsequent data collection for other questions (Adams et al., 2019).

Reviewer: 2

Reviewer Name: Assem M. Khamis

Institution and Country: Hull York Medical School, University of Hull, UK

Please state any competing interests or state 'None declared': I have no financial conflicts of interest.

Please leave your comments for the authors below

The article covers a critical topic -especially at the current time-: to examine the reporting of funding and conflicts of interest in the corresponding news of journal articles and press releases. The topic is crucial because the news and media formulate public knowledge and awareness about the new scientific evidence, and lack of reporting both COI and funding could affect how the public perceives the news and result in bias. The study is shedding the light on the importance of the industry funding not only for the scientific community but also for the lay audience. According to the study findings, there is a huge lack in reporting of funding sources and authors' conflicts of interest in the news and press releases corresponding to the published journal articles.

Major revisions:

Page 4 – 7: general comments on the 'Methods' section:

1- It would be helpful for the readers if subheadings were provided: study design, definitions, sources of documents, search methods, inclusion criteria, data extraction and coding, and data analysis.

Thank you for the suggestion. We have added such headings.

2- I think there is a repetition in reporting on the numbers of articles, press, and news reported in the methods and results section. It might be better to keep it just in the results section except for the coding reliability statistics.

Thanks for pointing this out. We have removed the lists of N from the analysis section of results, since they reappear soon after in Results.

3- The authors did an excellent job in coding in duplicate and independently to minimize data extraction errors.

4- A flow diagram for the included documents will be helpful to track the documents' selection process and analyses. Similar to the previous study with the documents database https://eur03.safelinks.protection.outlook.com/?url=https%3A%2F%2Fwww.bmj.com%2Fcontent%2F349%2Fbmj.g7015&data=02%7C01%7CSumnerP%40cardiff.ac.uk%7Cced9d421a3b446799f5008d865f6d439%7Cbdb74b3095684856bdbf06759778fcbc%7C1%7C0%7C637371458089310343&sdata=sef1LEl6l%2Bkt26AMpwmWUDFdmaec%2BBu2IRPX9obf4gk%3D&reserve_d=0

Thank you for the suggestion. We have added a flow diagram as figure 1.

Page 9 – 11: general comments on the ‘Discussion’ section:

1- It would be helpful for the readers if subheadings were provided: summary of findings, strengths and limitations, comparison to other studies, impact on future research and practices in news and press releases, and conclusions.

Thank you for the suggestion. We have added subheadings.

2- ‘what’s next?’ it would be great if the authors could reflect how their study findings could help in improving the press and news environment to report on COI and funding in scientific related materials. Should these media sources acquire COI and funding policies to report on the scientific findings? Should the universities have their policies for the press releases?

We include the following suggestions in the discussion, and have made this more explicit by moving that paragraph up in the Discussion: *It could be beneficial if press offices at medical journals, funders and academic institutions were to routinely highlight funding and disclosures in their press releases. One way to routinely operationalize this approach would be to add standard headers in press releases for funding and conflicts of interest as is done in many medical journals. Formal testing of alternate content and formats would inform the creation of more effective press releases. If press releases were made openly available and linked to publications for peers to scrutinize, this might remind authors to declare their conflicts of interest*

Page 9, line 164: the paragraph started with “We additionally observed that university press releases...” the authors introduced new information that has no prior been mentioned in the methods or the results section. Despite the thoughtful reflection about this observation, the authors are free to provide data on it in the results section to allow the readers to reflect on these numbers or remove this paragraph from the discussion as it will not affect the discussion flow.

Apologies this was unclear. The information was included in the second paragraph of results: *In the larger sample (2011 cohort) we could divide press releases issued by universities from those issued by journals; the universities were more likely to mention a funding source (59% vs 5%; absolute difference=54%, 95% CI: 49% to 59%).* We agree it was not a key part of the results, and it does not deserve such emphasis in the second paragraph of Discussion. We have moved and shortened the discussion paragraph.

Minor revisions:

Page 2, line 5: Objectives: there is a repetition “in”

Fixed.

Table 1 is a little bit crowded with numbers. I would suggest doing the following:

1- ‘n’= 996 should ‘N’ for the baseline denominators for each year

We have amended this

2- I think there is no need to mention ‘N’ in every statistic and it would be less crowded if n (%) if reported. For example: “Any industry” 169 (17.0)

We agree table 1 is crowded and have attempted to make this change, and those below. However we found that although the table then looked cleaner it became harder to interpret. So we put it back again to the format in the original submission. The reasons are twofold: The reader is primarily interested in the % prevalence, but wants to understand how the % was calculated – since, as point 3 points out, there are often two or more ways (e.g. % of reports in all university press releases, 253/426, or % of university reports in all reports, 253/284). The clearest way to disambiguate this is to put the calculation right next to the %; otherwise the reader does not know if to look up the column or across the row to find the appropriate denominator. Second, the denominator can change within columns and within rows, so there is no simple consistent method for labelling the rows and columns with their denominators.

3- In the row of “All press releases”: it is better to stick to column percentages for university and journal instead of row percentages. For example: 253/284 or 253/996. And ‘n’ for university and journal can be added beside them as follow:

University (n= 426)

Journal (n= 570)

As outlined above, we attempted to follow this advice, but we think it adds a possible ambiguity for the reader if the denominator is not right next to the %. In terms of which % to give, we think most readers are probably more interested in the % of reports in all university press releases, 253/426 (which stands on its own, regardless of what happens in Journals), rather than % of university reports in all reports, 253/284 (which depends also on what Journals do). But as long as it is clear what we have done, readers can calculate the other % for themselves.

4- In case of new denominators, N can be reported against the main variable.

For example: “Authors conflict of interest disclosed” n= 268

As above.

Reviewer: 3

Reviewer Name: Andreas Lundh

Institution and Country: Centre for Evidence-Based Medicine Odense (CEBMO), Odense University Hospital, Denmark

Please state any competing interests or state ‘None declared’: None

Please leave your comments for the authors below

General comments

The authors report on a study of 1250 press releases on scientific articles and corresponding uptake by news media. The authors have previously used this sample to study exaggeration in news and the focus of the current study is whether study funding and authors’ conflicts of interest is reported in news. The authors find that study funding is often not reported and reporting of funding in press releases is associated with corresponding reporting in news. In addition, authors’ conflicts of interest are rarely reported in both press releases and news media. The study is both interesting and generally well reported. However, I have some comments that the authors may consider.

Major compulsory revision

None.

Minor compulsory revision

Title and abstract The title describes the study as a “retrospective observational study” and the abstract as a “retrospective quantitative content analysis”. I suggest the authors describe the study using the same terminology throughout the paper and “retrospective observational study” seems somewhat too vague. Since the authors report on two cohorts of journal articles with corresponding

press releases and follows these articles over time to see how they are cited in news I think they could also call this a cohort study of two cohorts.

Thank you for this advice. To combine these suggestions we have opted for 'retrospective content analysis with two cohorts.'

p6 Methods A protocol is mentioned in the methods section. I suggest that the authors add the protocol as supplementary information or link to a public available version.

Good point. The data and protocols for the databases and the data extraction for this study are available online, and we should have included the links. We do now.

p6 Methods Both cohorts include press releases sampled from both journals and universities. However, a single research articles may have multiple press releases e.g. one from the journal and one from each of the co-authors' institutions. It is not clear from the manuscript if the same journal article could be included multiple times with different press releases or if this was not the case how the authors decided which press release to use in case of multiple press releases identified for a single article.

This is an astute observation. This intersection was not common (presumably because there are just so many journals universities submit to, and so many places each journal gets submissions from), but when it occurred we only included one press release, randomly selected.

p7 para 3 line 1-4 + Table 1 It seems that the category "industry-funded" was used if a study had received any industry funding, despite for example NIH funding being mentioned first. However, according to table 1 in the 2015 cohort of all studies the total is 254, but according to funding categories the sum is 268 studies and for the industry subsample the total is 134 and the sum is 141. The authors should please clarify this discrepancy.

A study could belong to more than one category. In the example of NIH plus industry funding, it would belong to both funding categories. We have clarified this in the methods and legend.

p8 para 2 line 1-5 It seems that the authors only coded whether a study with reported funding or conflicts of interest also had funding or conflicts of interest reported in press releases and news and not whether the subsequent reporting adequately reflected the information in the journal article. For example, the journal article might mention multiple funding sources and authors' conflicts of interest, but only some of these may be reported in press releases and news. The authors should consider discussing this as part of the study limitations.

This is a good point, and we have added a mention in limitations. We did actually initially code whether all, some or none of the sources / COI were reported, but we found that it was too complicated to make an intuitive analysis plan for this 3-level distinction, so we pulled back to just 'yes or no' overall (collapsing some and all), and separately 'yes or no' for industry funding. Given the rarity of reporting in news, we do not think it would change the overall message.

p9 para 2 line 6 It is unclear how the authors converted odds ratios to relative risks.

Sorry, this was a legacy statement from previous drafts. Sumner and Chambers reported OR in previous papers, while Schwartz and Woloshin reported RR. In the first draft for this collaboration, Lisa converted our OR to RR for added readability, but since then we have calculated RR directly from the estimated risks from the GEE output. We have amended the document.

Discretionary revision

Abstract objectives and main manuscript The authors describe one of their objectives as “whether disclosure in news is associated with disclosure in press releases”. To better fit with the possible causal nature and temporal relationship I suggest they turn the description of the association around i.e. whether disclosure in press releases is associated with disclosure in news.

Interestingly we read the inference the other way round: with the outcome of interest first and the possible causal factor second (e.g. climate change is associated with CO2 emissions), but the reader is a better judge, so we have changed it.

Abstract results The authors write that the results from the 2011 and 2015 cohorts were similar. However, funding was reported in 52% of 2015 press releases compared with 29% of 2011 press releases. Since this is a result I suggest the authors do not make any qualitative interpretation and delete “similar”.

We have deleted the word similar. We used it because we did not want the reader to think we are making a direct contrast of these rates. More important is the broad reduction from around 90% in journals to 10% in news.

p5 para 3 line 7-8 The last sentence is somewhat unclear. Why not use “are associated with”?

Our document has different page numbers it seems, but we think this refers to the sentence ending ‘tracked with....’. We agree it was hard to parse and have changed it around and now use *associated with*.

p6 para 2 line 11 The authors mention that cohort 1 is from UK universities, but the geographical location of the universities in cohort 2 is not described.

Sorry for this omission. We have corrected it. they were also UK, and consisted the Russell Group plus additional Universities included in Adams et al. 2019

p7 para 2 line 8 It is unclear how the “largest” funding source was decided as specific monetary amounts are rarely reported in journal articles.

Sorry this was unclear. The assumed largest was the first mentioned (the ‘or’ meaning defined by, not alternatively). We have clarified by deleting ‘largest’.

p8 para 2 line 1-3 The coding of funding information from press releases and news is also mentioned in the section “funding – general” above so this seems somewhat like repetition.

We agree, and have changed the wording to clarify the point mentioned above about the coding being yes or no, and not including ‘some’.

p8 para 3 For cohort 1 it is not described what the authors did when coders disagreed.

Thanks for spotting; we have now included the statement: For coding disagreements, one answer was randomly selected.

p9 results I suggest also writing actual numbers in addition to percentages, e.g. 934 (94%) listed sources of funding.....

We have amended for the first sentences in results, but then refer the reader to figure 1, as it impairs reading to include the numbers for all the sentences.

p9 para 3-4 (heading) I suggest writing coi in full.

We have amended

p10 para 1 The authors describe that press releases rarely reported no conflicts of interest when none were reported in the journal article. However, the subsequent example is not related to authors’ conflicts of interest, but whether the study was independently conducted without funder involvement.

A study might be “independently” conducted by authors with personal financial conflicts of interest with the funder. I suggest the authors should provide another example.

The example was for the type of sentence we coded as a declaration of no conflict in press releases. Strictly, I agree there this is not quite the same thing, but since it was so rare we were inclusive. This was the only example. We have deleted the example altogether as it seems to misrepresent the data to provide a memorable example for the rarest of occurrences, while not providing examples for all other aspects of the data.

p11 para 2 line 5-8 I suggest moving this section from the principal finding to the study interpretation section below.

We think this refers to the sentences starting ‘Note that we do not infer anything from the fluctuations between years.’ We have moved them as suggested.

P13 para 2 line 12-16 I suggest using the term non-financial conflicts of interest or non-financial interest and consider citing the discussion concerning such interest by Bero and Grundy (Bero PLoS Biol 2016).

We have changed the wording and cited this work; thanks for pointing it out.

Table 1 In main text funding source category is internal/other, here it is internal.

Amended.

VERSION 2 – REVIEW

REVIEWER	Joel Lexchin York University Canada
REVIEW RETURNED	01-Nov-2020
GENERAL COMMENTS	The authors have dealt with my comments on the first draft.
REVIEWER	Assem M. Khamis Hull York Medical School, University of Hull
REVIEW RETURNED	14-Nov-2020
GENERAL COMMENTS	The authors addressed all the comments in the revised version. I have no further comments.